

# VPRM-CHINA: Using the Vegetation, Photosynthesis, and Respiration Model to partition contributions to CO₂ measurements in Northern China during the 2005-2009 growing seasons.

Archana Dayalu[1,2], J. William Munger[2], Steven C. Wofsy[1,2], Yuxuan Wang[3,4], Thomas Nehrkorn[5],
Yu Zhao[6], Michael B. McElroy[2], Chris Nielsen[2], Kristina Luus[7]

[1]Department of Earth and Planetary Sciences, Harvard University, Cambridge, 02138, United States of America
[2]School of Engineering and Applied Sciences, Harvard University, Cambridge, 02138, United States of America
[3]Department of Earth and Atmospheric Sciences, University of Houston, Houston, 77204, United States of America
[4]Department of Earth System Sciences, Tsinghua University, Beijing, 100084, People's Republic of China
[5]Atmospheric and Environmental Research, Inc., Lexington, 02421, United States of America
[6]School of the Environment, University of Nanjing, Nanjing, 210023, People's Republic of China
[7]Centre for Applied Data Analytics (CeADAR), Dublin 4, Ireland

*Correspondence to*: Archana Dayalu (adayalu@seas.harvard.edu)

**Abstract.** Accurately quantifying the spatiotemporal distribution of the biological component of CO₂ surface-atmosphere
exchange is necessary to improve top-down constraints on China's anthropogenic CO₂ emissions. We provide hourly fluxes
of CO₂ as Net Ecosystem Exchange (NEE; $\mu molCO_2m^{-2}s^{-1}$) on a 0.25°x0.25° grid by adapting the Vegetation,
Photosynthesis, and Respiration Model (VPRM) to the eastern half of China for the time period from 2005-2009; the
minimal empirical parameterization of the VPRM-CHINA makes it well-suited for inverse modeling approaches. This study
diverges from previous VPRM applications in that it is applied at large scale to China's ecosystems for the first time,
incorporating a novel processing framework not previously applied to existing VPRM versions. In addition, the VPRM-
CHINA model prescribes methods for addressing dual-cropping regions that have two separate growing season modes
applied to the same model gridcell. We evaluate the VPRM-CHINA performance during the growing season and compare to
other biospheric models. We calibrate the VPRM-CHINA with ChinaFlux and FluxNet data and scale up regionally using
Weather Research and Forecasting (WRF) Model v3.6.1 meteorology and MODIS surface reflectances. When combined
with an anthropogenic emissions model in a Lagrangian particle transport framework, we compare the ability of VPRM-
CHINA relative to an ensemble mean of global hourly flux models (NASA CMS) to reproduce observations made at a site in
Northern China. The measurements are heavily influenced by the Northern China administrative region. Modeled hourly
timeseries using vegetation fluxes prescribed by VPRM-CHINA exhibit low bias relative to measurements during the May-
September growing season. Compared to NASA CMS subset over the study region, VPRM-CHINA agrees significantly
better with measurements. NASA CMS consistently underestimates regional uptake in the growing season. We find that
during the peak growing season, when the heavily cropped North China Plain significantly influences measurements,
VPRM-CHINA models an CO₂ uptake signal comparable in magnitude to the modeled anthropogenic signal. In addition to
demonstrating efficacy as a low-bias prior for top-down CO₂ inventory optimization studies using ground-based



measurements, high spatiotemporal resolution models such as the VPRM are critical for interpreting retrievals from global $CO_2$ remote sensing platforms such as OCO-2 and OCO-3 (planned). Depending on the satellite time-of-day and season of crossover, efforts to interpret the relative contribution of the vegetation and anthropogenic components to the measured signal are critical in key emitting regions such as Northern China--where the magnitude of the vegetation $CO_2$ signal is

shown to be equivalent to the anthropogenic signal.

# 1 Introduction

In 2006, China surpassed the USA as the world's leading anthropogenic carbon dioxide ($CO_2$) emitter. China's contribution to world $CO_2$ emissions has been growing steadily and now constitutes approximately 26% of the world total, compared to the USA's 17%, accounting for 60% of the overall growth in global $CO_2$ emissions over the past 15 years (EIA, 2017). China

and the USA made an historic joint announcement on national carbon commitments in November 2014, an unprecedented form of political coordination by the two countries to advance United Nations climate negotiations. In addition, China pledged at the 2015 UN climate summit in Paris to peak carbon emissions by 2030; in March 2016, China released its 13[th] Five-Year Plan to strengthen its strategies to achieve its emission targets (Tollefson, 2016). An accurate assessment of $CO_2$ fluxes within China is not only a critical advance in testing the bottom-up emission inventories that provide the baselines for

setting such policy commitments and measuring progress, it also broadens understanding of the country's contributions to climate change beyond the sources conventionally targeted for control. Eventually such observation-based assessments might be formally integrated into regulatory processes to strengthen baselines, to widen the scope of control, and to assess policy progress and compliance.

Good prior estimates of the spatiotemporal structure of $CO_2$ surface exchanges are needed to reduce the uncertainty in top-down optimizations where atmospheric observations are used as a constraint to improve bottom-up flux inventories. As a first step towards evaluating China's anthropogenic emission inventories on an intra-annual basis, it is necessary to also model vegetation contributions to the $CO_2$ signal during the growing season. Previous studies (Wang et al., 2010) relied on $CO_2$ to CO ratios to estimate annual anthropogenic $CO_2$ emission enhancements from winter-time observational data alone.

The large diurnal $CO_2$ uptake and emission vegetation signal in the growing season complicated modeling the anthropogenic $CO_2$ signal during these times of the year. Seasonal variations in anthropogenic emissions patterns from both shifts in atmospheric transport and emission sources themselves are therefore insufficiently captured when dormant season observations alone are used to estimate annual emissions.

This study adapts the Vegetation, Photosynthesis, and Respiration Model (VPRM; Mahadevan et al., 2008) to model $CO_2$ vegetation fluxes on an hourly basis on a 0.25°x0.25° grid from 2005-2009. VPRM-CHINA is empirically driven with very low dimensional parameterizations, making it particularly suitable as a biogenic $CO_2$ flux prior model for top-down inverse



model analysis of China's emissions. We demonstrate its validity as a prior for use in atmospheric inversions that constrain anthropogenic $CO_2$ emissions on inter- and intra-annual scales by comparison to hourly observations at a site in Miyun, China, 100km northeast of the Beijing urban center.

## 2 Methods

As described in detail by Mahadevan et al. (2008), modeled $CO_2$ vegetation fluxes (Net Ecosystem Exchange; NEE) from VPRM-CHINA are calibrated against observed $CO_2$ fluxes for each dominant vegetation class represented in the study domain. $CO_2$ exchange is dependent on ecosystem temperature and sunlight, driven here by high-resolution meteorology from the Weather Research and Forecasting (WRFv3.6.1) model (http://wrf-model.org). The photosynthetic capacity of ecosystems is controlled by vegetation greenness and water availability, and these factors are obtained from MODIS remote-sensing (https://lpdaac.usgs.gov) land cover and surface reflectance data sets. We define uptake and release of carbon relative to the atmosphere such that the photosynthesis (Gross Primary Productivity; GPP) term is negative (representing $CO_2$ uptake from the atmosphere) and the ecosystem respiration ($R_{eco}$) term is positive (representing $CO_2$ release to the atmosphere). Modeled $CO_2$ in the growing season is ultimately evaluated against hourly averaged observations collected from 2005 to 2009.

This study diverges from previous VPRM applications in three main ways: (1) the VPRM-CHINA is applied at large scale to China's ecosystems for the first time; (2) the scaling involves a novel processing framework not previously applied to existing versions of the VPRM; and (3) the VPRM-CHINA model prescribes methods for addressing winter wheat and corn dual-cropping regions that have two growing season modes for the same pixels.

Data processing software used in this study are the MODIS Reprojection Tool (MRT) Release 4.1 (https://lpdaac.usgs.gov/tools/modis_reprojection_tool); the R program for statistical computing (Rv3.2.0, https://www.r-project.org/); and NCAR Command Language (NCLv6.2.1; http://dx.doi.org/10.5065/D6WD3XH5).

We begin this section with an overview of the observational record used in this study (Sect. 2.1). Sect. 2.2 presents details of VPRM-CHINA, including processing of model drivers and model calibration. Sect. 2.3 introduces the anthropogenic emissions inventory used in this study. Sect. 2.4 concludes the methods section, summarizing the derivation of the modeled $CO_2$ time-series.

### 2.1 $CO_2$ Observations

We evaluate performance of the VPRM-CHINA during the growing season using five years (2005-2009) of continuous hourly $CO_2$ observations (LI-COR Biosciences Li-7000). The site (Miyun; 40°29'N, 116°46.45'E) is in a rural area in





Northern China, 100km northeast of the Beijing urban center. Miyun is located south of the foothills of the Yan mountains, and is influenced by clean continental air from the northwest and polluted urban air from the southwest. The vicinity is primarily grasslands, croplands, and mixed temperate forest. Further descriptions of the site and details of the instrumentation of the $CO_2$ observations are in Wang et al. (2010).

## 2.2 VPRM-CHINA

### 2.2.1 Model Overview

We follow the general model framework established by Mahadevan et al. (2008) to construct hourly $CO_2$ *NEE* estimates on a $0.25°$x$0.25°$ grid over the time period from 2005 to 2009. The hourly *NEE* is modeled as a function of temperature sensitivity ($T_{scale}$), phenology ($P_{scale}$), water stress ($W_{scale}$), photosynthetically active radiation (*PAR*) and the Enhanced Vegetation Index

(*EVI*). As shown in (1) below, modeled *NEE* is partitioned into *GPP* (the first parenthesized term) and $R_{eco}$ (the second parenthesized term):

$$NEE = -\left(\lambda \times T_{scale} \times P_{scale} \times W_{scale} \times \frac{1}{1+\left(\frac{PAR}{PAR_0}\right)} \times EVI \times PAR\right) + (\alpha \times T + \beta) \qquad (1)$$

The parameters $\lambda$, $\alpha$, $\beta$ and $PAR_0$ are empirically adjusted based on calibrations against observed *NEE* from eddy flux data

for each MODIS vegetation class based on the International Geosphere-Biosphere Programme (IGBP; MCD12Q1) in the domain. $PAR_0$ represents the half-saturation value of photosynthetically active radiation. In addition, we set a minimum temperature threshold for each vegetation class, $T=T_{low}$ ($1 \le T_{low} \le 5°C$) prescribing a baseline of soil respiration at very low air temperatures (Hilton et al., 2013; Mahadevan et al. 2008). $T_{low}$ is derived from fits to site-level data; see Sect. 2.2.4 for details.

The temperature sensitivity is defined as below, where $T_{min}$, $T_{max}$, and $T_{opt}$ represent minimum, maximum, and optimal temperatures for photosynthesis respectively and are set at literature values for each vegetation class. Temperature $T$ is the hourly averaged 10-min surface temperature output from the WRFv3.6.1 meteorological model (Sect. 2.2.3). With the exception of winter wheat, we use the same $T_{min}$, $T_{max}$, and $T_{opt}$ for each ecosystem type in our domain as in the Mahadevan et

al. (2008) North America study. The similarity of latitudes for each ecosystem type between the Mahadevan et al. (2008) study and this study makes this an appropriate approximation. The only ecosystem category in our domain not represented by Mahadevan et al. (2008) is winter wheat; as such we set our winter wheat $T_{min} = 0C$ and $T_{opt} = 20C$ , the lower values relative to other crop types reflecting the lower temperatures of the winter wheat growing season (Acevedo et al. 2002).

$$T_{scale} = \frac{(T-T_{min}) \times (T-T_{max})}{\left[(T-T_{min}) \times (T-T_{max}) - (T-T_{opt})^2\right]} \qquad (2)$$




$P_{scale}$ and $W_{scale}$ are functions of the Land Surface Water Index (*LSWI*). Both *LSWI* and the Enhanced Vegetation Index (*EVI*) are derived from the MODIS surface reflectance data set (MOD09A1) as in (3a) and (3b), where the surface reflectance bands used are the red band *($\rho_{red}$*, band 1); near infrared band ($\rho_{nir}$, band 2); blue band ($\rho_{blue}$, band 3); and the shortwave infrared band ($\rho_{swir}$, band 6).

$$LSWI = \frac{\rho_{nir} - \rho_{swir}}{\rho_{nir} + \rho_{swir}} \tag{3a}$$

$$EVI = 2.5 \times \frac{\rho_{nir} - \rho_{red}}{\rho_{nir} + (6 \times \rho_{red} - 7.5 \times \rho_{blue}) + 1} \tag{3b}$$

The water stress parameter $W_{scale}$ and phenology parameter $P_{scale}$ are defined consistent with Mahadevan et al. (2008) and are shown in (4) and (5) below. Ecosystem timing events are determined either manually (cropland classes for each degree latitude from 32N to 38N) or from the MODIS timing product (MOD12Q2; all other vegetation classes and croplands in other latitude zones):

$$W_{scale} = \frac{1 + LSWI}{1 + LSWI_{max}} \tag{4a}$$

$P_{scale}$ is set to 0 for water, snow and ice, and unclassified pixels at all times. For evergreen classes at all times and other vegetation classes at maximum greenness we set $P_{scale}$ to 1. We represent phenology as a fraction of *LSWI* for non-evergreen vegetation classes from (1) onset greenness increase to greenness maximum, and (2) onset greenness decrease to greenness minimum:

$$P_{scale} = \frac{1 + LSWI}{2} \tag{4b}$$

Ecosystem timing dates for selection of the appropriate $P_{scale}$ parameterization for each pixel were obtained from the MOD12Q2 phenology product, detailed in Sect. 2.2.2.

### 2.2.2 Satellite Data Processing

We use tiles from three MODIS products on a 500m sinusoidal grid to model *GPP*: 8-day average MOD09A1 surface reflectance bands 2, 6, 1 and 3 representing the near IR, short wave IR, red, and blue regions respectively; annual MCD12Q1 land use categories based on IGBP land classifications; and annual MOD12Q2 ecosystem timing dates. We do not include MODIS surface reflectance data from the Aqua satellite due to failure of a majority of band 6 detectors after launch, which affected availability of high-quality data during the time period investigated in this study. All datasets were downloaded

using the Reverb tool in NASA's Earth Observing System Data and Information System (http://reverb.echo.nasa.gov/). We





then used MRT to (1) mosaic tiles associated with our spatial domain and (2) reproject to a WGS84 datum Geographic Coordinate system on a 500m grid.

The dominant IGBP ecosystem types represented in the domain are effectively constant over the 5-year study period.

Together, water and the major photosynthesizing land classes constitute 98% of the domain (Figure 1). We follow the Mahadevan et al. (2008) convention where (1) we set the *NEE* of water, urban/built, and snow/ice to zero; and (2) group together (i) savannas and woody savannas; (ii) grasslands, croplands & natural mosaic, and barren & sparse; (iii) deciduous needle-leaf and deciduous broadleaf. The remaining non-dominant vegetation classes not represented in the above (evergreen needle-leaf, closed & open shrublands, permanent wetlands) collectively constitute <1.5% of the total land area and therefore

do not appreciably affect the carbon fluxes in the domain. Furthermore, closed and open shrubland ecosystems were found by Mahadevan et al (2008) to be outside of the scope of the VPRM-CHINA due to the inability to adequately capture the influence of inorganic soil carbon pools on observed carbon dioxide fluxes. Pixels corresponding to these ecosystem types have *NEE* values set to missing.

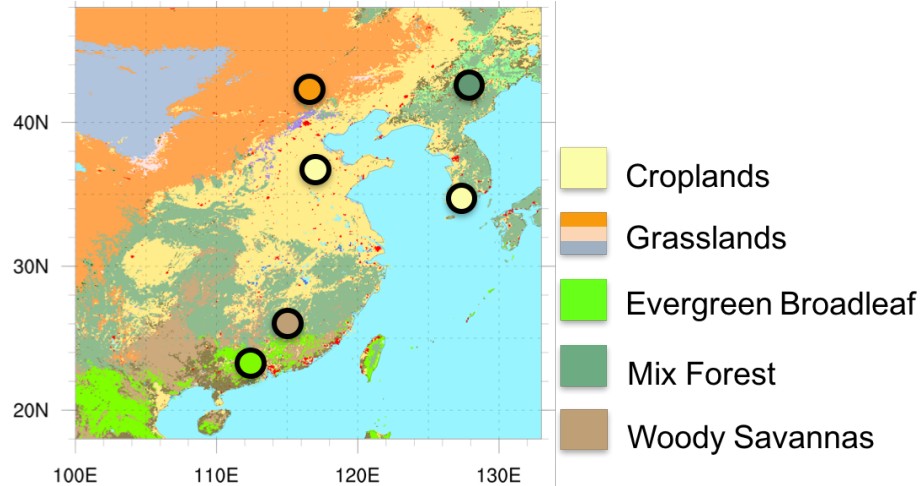

**Figure 1. Dominant IGBP categories from MOD12Q1 data over study spatial and temporal domain. Circled points represent approximate location and IGBP class of eddy flux sites used in VPRM-CHINA calibration.**

The reflectance data was quality filtered in Rv3.0.2 to accept only the highest quality data under clear sky conditions. Inconsistencies in the internal snow-cover flags made it necessary to manually filter erroneous reflectance values due to snow rather than ecosystem photosynthetic activity.

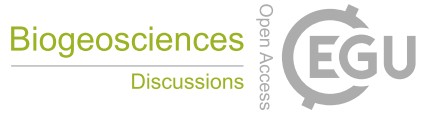

NCLv6.2.1 (NCL) was used for higher-level data processing. All missing values resulting from higher-level quality filtering steps were interpolated using NCL's Poisson grid filling algorithm and then used to calculate *EVI* and *LSWI*. *EVI* was further filtered to keep only values within a valid range (0 to 1). *LSWI*, driven by Mode 1 MCD12Q2 ecosystem timing dates (or manually selected timing dates for the 32N to 38N latitude belt), was used to calculate phenology cycles parameterized by

5   $P_{scale}$ and water stress parameterized by $W_{scale}$. The quality filtering of the ecosystem timing dates was done manually in NCL as there are known issues with the MCD12Q2 internal quality flags (https://www.bu.edu/lcsc/files/2012/08/MCD12Q2_UserGuide.pdf). Therefore, quality filtering of MODIS ecosystem timing dates was limited in scope to (1) removing anachronistic dates represented by instances where, for a given pixel, ecosystem times were not in chronological order, and (2) where a given pixel's date was outside of 1-σ of the mean for the ecosystem

10   class represented by that pixel for its latitude band.

The second step of MCD12Q2 quality filtering was not conducted for cropland classes in the 32N-38N latitude band. Croplands between 32N and 38N located within the North China Plain have a high prevalence of winter wheat/corn dual-cropping zones, where winter wheat dominates a cropland site in the spring months and corn dominates in the summer

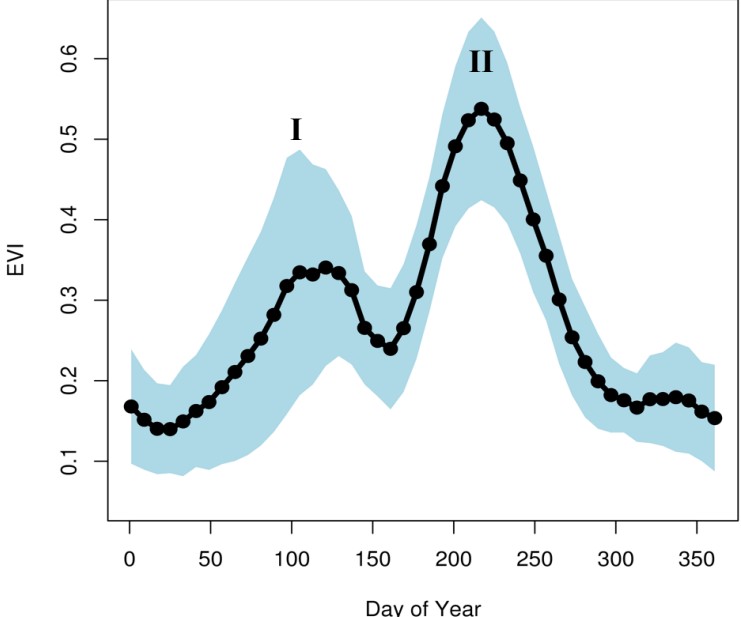

**Figure 2**. **Average *EVI* in IGBP cropland class from 32N to 38N. Bimodal peak represents (I) winter wheat spring emergence and (II) corn summer emergence. Shaded blue region represents 1-σ of spatial and temporal average.**



months. Considerably different ecosystem timing dates can occur for these dual croplands at the same latitude band and pixel due to the bimodality of the phenology. We subdivide cropland classes spatially and temporally based on analysis of average annual *EVI* for each degree latitude from 30N to 40N and USDA crop maps of major cropping regions of China (USDA, 2016). We designate all cropland classes south of 32N as rice; all cropland classes between 32N and 38N as winter
wheat/corn dual croplands, and all croplands north of 38N as corn. We further justify the designation of the 32N-38N cropland classes as winter wheat/corn by noting a distinct bimodal pattern of average *EVI* for that region (Fig. 2), similar to Yan et al. (2009). For the dual croplands only, we manually set for each of the two crop modes a multi-year (2005-2009) mean of phase timings for each degree latitude from 32N to 38N, obtained from *EVI* averaged across all cropland pixels at the respective degree latitude.

### 2.2.3 WRF Temperature and Radiation Fields

Hourly averaged surface temperature (T2) and downward shortwave radiation (SWDOWN) fields were derived from 10-minute WRF output. The WRF model was initialized with NCEP FNL $1^{\circ}x1^{\circ}$ resolution reanalysis data (NCEP, 2000) and run for independent 24h periods with three domains spanning the study temporal domain, excluding a 6h spinup time
according to practice established by Nehrkorn et al. (2010) and Hegarty et al. (2013). We employ continuous nudging in the outer domain only and we do not nudge any fields within the planetary boundary layer (PBL). The Yonsei University (YSU) PBL scheme is employed. The WRF model is well known to have excess shortwave radiation compared to observations, primarily caused by misrepresentation of clouds and their radiative effects (e.g., Ruiz-Arias et al., 2016). The RRTMG scheme employed in this study includes a method for random cloud overlap in a gridcell (WRF-ARW, 2014). Additional
improvements to the treatment of short-wave radiation have been made in more recent versions of WRF than used in this study (Jimenez et al., 2016). Here we will apply an empirical correction to reduce the bias in total incoming shortwave radiation.

The T2 and SWDOWN fields for the outer domain (27km gridcell resolution) are averaged to hourly intervals and then
regridded to the same coordinate system as the processed MODIS products using NCL's ESMF bilinear interpolation tool. *PAR* is very closely correlated with shortwave radiation, where SWDOWN $\approx 0.505 PAR$ (Mahadevan et al., 2008).

We quantify the high bias for WRF SWDOWN, and therefore *PAR*, for our specific study region by comparison of modeled *PAR* to measured *PAR* at eddy flux sites in the domain and scale our modeled *PAR* accordingly, by season, using the
aggregated *PAR* across five eddy flux sites.





### 2.2.4 VPRM-CHINA Calibration

Hilton et al. (2013) highlight the importance of tailoring *NEE* parameter estimates in VPRM to the specific region being studied. As such, we obtain VPRM-CHINA parameters by calibrating the major ecosystem types to representative ecosystem eddy flux data in the domain (Table 1). We use unfilled eddy flux data from the Fluxnet 2015 database

(http://fluxnet.fluxdata.org/data/fluxnet2015-dataset), ChinaFlux (www.chinaflux.org), and the Fluxnet LaThuile synthesis dataset (http://fluxnet.fluxdata.org/data/la-thuile-dataset/) to calibrate our modeled *NEE* output for each of our dominant ecosystem types. We used the average of up to nine model pixels of the same IGBP class: *NEE* from the 500m gridcell nearest to the eddy flux observation site and the surrounding 8 pixels. All observational data were hourly averaged from the original half-hour resolution data sets, and were filtered for sufficiently high frictional velocity, *u\**, to ensure well-developed

turbulence at canopy level prior to calibrating the VPRM-CHINA at each dominant ecosystem type (Goulden et al. 1996).

**Table 1. Calibration & Validation site information. *Site (year) used as validation.**

| Site Name (abbrev) | Location/ Site Elev | Ecosystem Type (IGBP Class; %domain) | Data Year | Data Source |
|---|---|---|---|---|
| Changbaishan (CN-Cha) | 42.40N, 128.1E/ 751.0 masl | Mixed Forests (5; 11%) | 2005 | Fluxnet— FLUXNET2015 |
| Dinghushan (CN-Din) | 23.17N, 112.5E/ 218.0 masl | Evergreen Broadleaf (2; 4.4%) | 2005 | Fluxnet— FLUXNET2015 |
| Duolun Grassland (CN-Du2) | 42.05N,116.3E/ 1333 masl | Grasslands (10; 21%) | 2007, 2008* | Fluxnet— FLUXNET2015 |
| Haenam (KR-Hae) | 34.55N,126.6E/ 13.74 masl | Rice Croplands (12; Croplands total: 14%) | 2006* | Fluxnet— Lathuile |
| Qianyanzhao (CN-Qia) | 26.73N, 115.1E/ 79 masl | Woody Savannas (8; 5.6%) | 2005 | Fluxnet— FLUXNET2015 |
| Yucheng (CN-Yuc) | 36.95N, 116.6E/ 12 masl | Winter Wheat/Corn Croplands (12; Croplands total: 14%) | 2005 | ChinaFlux, Yu et al., 2006 Yu et al., 2013 |

We fit *NEE* represented in (1) to observations of *NEE* at each eddy flux site using a non-linear least squares fit (NLS; Gauss–Newton algorithm) in Rv3.2.0. We fit against the subset of non-missing observations and use site-level measurements of air

temperature and *PAR*. Prior to fitting, we set air temperatures to the respective site $T_{low}$ for instances where the measured air

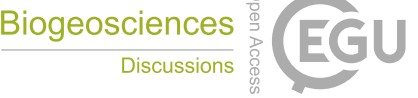

temperature falls below that threshold. Initial inputs of λ, α, and $PAR_0$ to the NLS algorithm are based on results from the respective IGBP ecosystem classes in Mahadevan et al. (2008) and Hilton et al. (2013). β is always initially set to 0.

With the exception of dual-cropped croplands, all VPRM-CHINA *NEE* parameters are obtained from fitting to observations for the whole year. For dual-cropped croplands, we fit the winter wheat and corn modes separately; for rice croplands, used
as a validation site only, we use WRF-derived and corrected *PAR*. With the exception of grasslands, there was insufficient observational data to provide both calibration data and validation data by either a different year in the same site or a different site of the same ecosystem type. In addition, the grassland site used in this study has a history of disturbance. The special cases of grasslands and croplands are discussed further, below.

Within the cropland IGBP ecosystem class we are restricted by availability of observational eddy flux data. Based on USDA agricultural maps we consider corn, winter wheat, and rice as a first order approximation of major croplands influencing $CO_2$ exchange in the study domain (USDA, 2016). We use 2005 ecosystem data from ChinaFlux site CN-Yuc to calibrate winter wheat/corn dual croplands. Winter wheat and corn parameters were fit to the observational subsets corresponding to times of the year where these crops are prevalent. The winter wheat seasonal subset was defined using dates earlier than July 1, 2005
and the corn seasonal subset was defined using dates on or after July 1, 2005. For corn, the CN-Yuc dataset was manually corrected for a data entry error that began in July of 2005 and lasted through the end of the year. Erroneous instances were flagged where the measured maximum of diurnal *NEE* uptake lagged measured and modeled *PAR* by two hours. The NLS fit of *NEE* was then performed as previously described on this offset-corrected data. Cropland pixels south of 32N were designated as rice and use 2006 ecosystem data from KR-Hae, a rice paddy site in South Korea, to validate rice cropland
parameters. KR-Hae dataset contained significant data gaps (43% of data) and could not reliably be used for calibration with the NLS method. Instead we use grassland parameters from Hilton et al. (2013) to represent rice cropland parameters and use KR-Hae *NEE* observations to validate. In addition, KR-Hae did not include *PAR* observations; therefore we used WRF-derived *PAR*, scaled by seasonal scaling factors in calculations of modeled *NEE*. Due to the large distances of rice croplands from the Miyun receptor, errors in rice parameterization as a result of this approach are not expected to appreciably affect the
final $CO_2$ concentration estimates at the receptor.

Grasslands were calibrated using CN-Du2 site data from 2007 and validated using CN-Du2 site data from 2008. The grassland is located in Inner Mongolia and faces low wintertime air temperatures. We find the eddy flux data was collected via an open path system. Insufficient WPL correction used to correct for air density changes (from heating of sensors for
example, during the winter time) resulted in possible $CO_2$ uptake artifacts during the dormant season. Following the convention of Mahadevan et al. (2008) we do not set a $T_{low}$ for grasslands. In addition, the CN-Du2 site represents a





transition site (grassland steppe to heavily grazed agricultural) that could impact its ability to represent undisturbed grasslands in the study domain (Zhang et al. 2007; Zhang et al. 2008).

### 2.3 Anthropogenic $CO_2$ Emissions Inventory

We use the annual anthropogenic $CO_2$ emissions inventories produced by the Harvard-China Project (ZHAO; Zhao et al.
2012) to represent the anthropogenic contribution to the observed $CO_2$ during the growing season. The ZHAO annual inventories are the first statistically rigorous bottom-up anthropogenic $CO_2$ inventories for China and integrate data from field studies specific to China's energy processes, technologies, and activity factors with an increased reliance on provincial-level data relative to national level data. The ZHAO inventories provide emissions in $GgCO_2$ on a 0.25°x0.25° grid for 2005 and 2009; for 2006-2008 we spatially allocate the total estimated emissions based on the percentage contribution of each
gridcell averaged between 2005 and 2009. In addition, we do not apply any temporal activity factors such as time of day or season of year changes in activity intensity. We therefore directly scale annual gridcell emissions (as $Gg$ $CO_2$) to $\mu molCO_2$ $m^{-2}s^{-1}$.

### 2.4 WRF-STILT: Derivation of Modeled Hourly $CO_2$ Timeseries

Each hourly measurement is modeled as advected background air uninfluenced by the study domain plus the integrated
effects of $CO_2$ sources (enhancements relative to background) and sinks (depletion relative to background) from surface processes in the study domain over a specified time period (here, up to seven days back from time of measurement).

We quantify the influence of surface processes using the Stochastic Time-Inverted Lagrangian Transport Model (STILT; Lin et al., 2003), an adjoint that computes surface "footprints" (ppm $\mu mol^{-1}m^{-2}s^{-1}$) for each measurement hour up to 168 hours
back from the hour of measurement. An ensemble of 500 hypothetical particles is sent back from the measurement point, driven by WRF meteorology (Nehrkorn et al., 2010). Each footprint ultimately represents the sensitivity of downwind concentration measurements made at a certain hour to upwind surface fluxes. We merge these hourly footprint maps with the vegetation and anthropogenic flux maps pertaining to the appropriate hour (constant in the case of the anthropogenic fluxes). We can then separately obtain the total anthropogenic enhancement and the vegetation enhancement (or depletion) to the
background signal at each measurement hour.

Background concentrations of $CO_2$ are estimated using NOAA CarbonTracker CT2015 (NOAA, 2016) provided on a 3-hourly 3°x2° global grid with 25 vertical levels, using a method similar to Karion et al. (2016). A background $CO_2$ concentration for each hour is calculated as follows: each STILT particle is assigned a background value at the end of its
back trajectory. A nearest neighbor approach selects the appropriate $CO_2$ background concentration based on the particle's end time, latitude, longitude, and altitude. A concentration is only considered to truly represent "background" if at least one



of the following criteria are met (i) the end point of a particle's back trajectory is the edge of the study's spatial domain; or (ii) if the particle has not reached the edges of the domain, its altitude must be greater than or equal to 3000 masl. A modeled hourly concentration value is then considered valid and included in the analysis if at least 75% of particles satisfy the background selection criteria; the valid background values are then averaged to provide one concentration for each

5 measurement hour. The selection criteria ensure that surface processes in the study domain did not interfere with "background" air during the time period of relevance to the analysis.



## 3 Results and Discussion

The final VPRM-CHINA product is hourly estimates of *NEE* on a 0.25°x0.25° grid. Figure 3 displays seasonal averages of hourly VPRM-CHINA *NEE* over the entire study time period.

5     The region of high springtime productivity in the North China Plain represents the manually prescribed winter wheat mode. Mixed forests at southern latitudes are given the same VPRM-CHINA ecosystem parameters as their only calibration site is a high-latitude mixed forest (CN-Cha; Table 1). This likely leads to an underestimate of mixed forest ecosystem productivity in the south as evidenced by zones of positive summertime mean *NEE* in southern mixed forest regions.

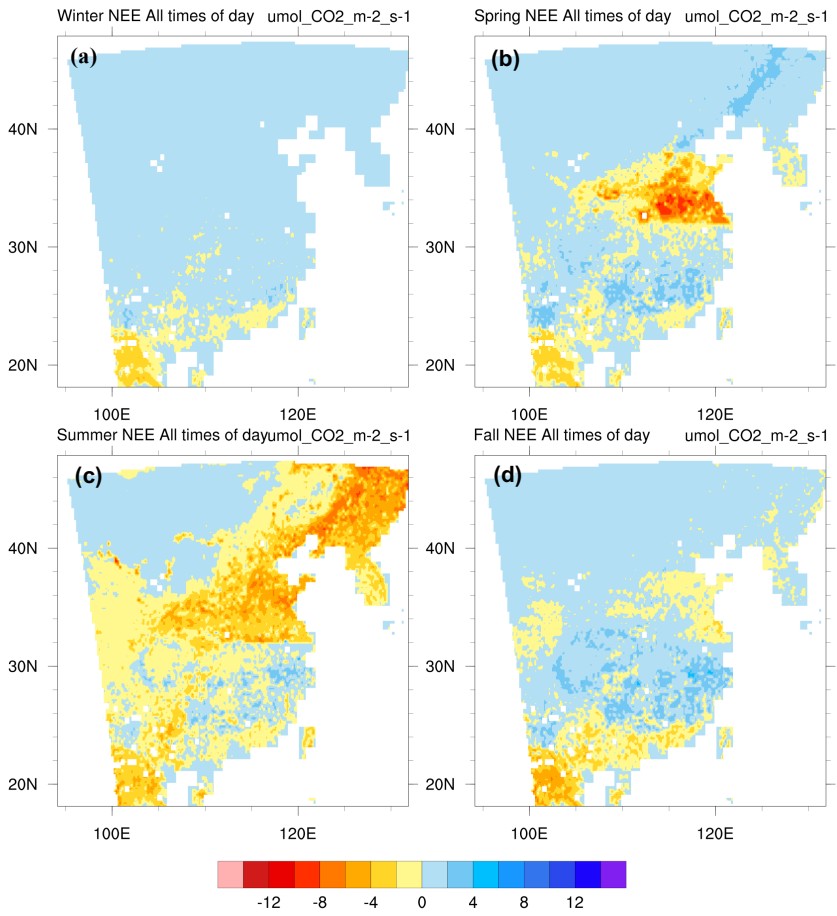

**Figure 3. Mean *NEE* ($\mu$mol $CO_2$ m$^{-2}$s$^{-1}$) averaged over all hours of day from 2005-2009 (a) DJF/Winter (b) MAM/Spring (c) JJA/Summer and (d) SON/Fall.**




We break down the results of our study as follows. In Sect. 3.1 we present results from comparison of *PAR* derived from WRF SWDOWN to *PAR* measured at the eddy flux stations and use these results to inform scaling of modeled *PAR* in the larger domain. In Sect. 3.2 we present results from calibrating the VPRM-CHINA to eddy flux station observations. In Sect.

5   3.3, we compare output from VPRM-CHINA to the NASA Carbon Monitoring System project (CMS; Fisher et al., (2010)). In Sect. 3.4, we compare the performance of VPRM-CHINA and CMS in an analysis of multi-year growing season contributions to $CO_2$ measured at the Miyun station. We conclude with Sect. 3.5 where we compare modeled estimates of annual carbon balance within the regions that are estimated as having the greatest influence on the observations.

**3.1 Seasonal Scaling Factors for Modeled PAR**

10   The WRF *PAR* bias exhibits seasonal variation, with the highest bias in winter (Fig. 4). We scale modeled *PAR* in Eq. (1) for each season with the derived seasonal scaling factors shown in Fig. 4b, obtained using ranged major axis fits to measured *PAR* (Legendre and Legendre, 1998). During the dormant season, zones where the $P_{scale}$ phenology term is set to 0 are unaffected by modeled *PAR* as the light-dependent portion of the *NEE* equation drops out. We find scaling *PAR* to be

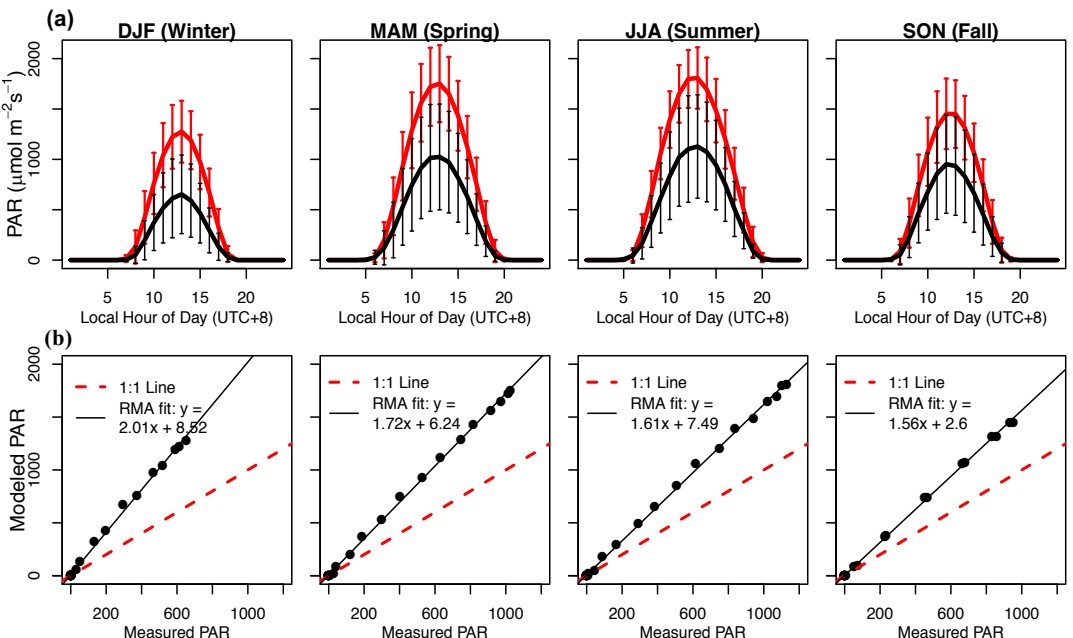

**Figure 4. Derivation of PAR seasonal scaling factors. (a) Aggregated mean modeled and measured PAR for each eddy flux calibration site by season; error bars are 1-σ. (b) Scaling factors for each season, based on fitting modeled to measured PAR using Ranged Major Axis regression.**



essential to capturing hourly processes influencing measured $CO_2$ during the growing season (Sect. 3.4). Failing to scale WRF-derived *PAR* with observations would result in a bias factor of 1.5 to 2, leading to an unrealistic overestimate of growing season hourly uptake and annual uptake of $CO_2$ in the domain.

### 3.2 VPRM-CHINA Calibration Results

Table 2 displays the results from calibration to eddy flux sites in the domain, and compares to results for similar ecosystem classes in North America (Hilton et al., 2013; Mahadevan et al., 2008). Table 3 summarizes the residual standard error (RSE) and the 1-σ values from the fits of each parameter. Figure 5 displays diurnal performance of the model relative to measurements by site during peak growing season and further partitions the model into the photosynthesis and respiration components. Monthly mean modeled vs. measured *NEE* is shown for all sites (calibration and validation) in Fig. 6.

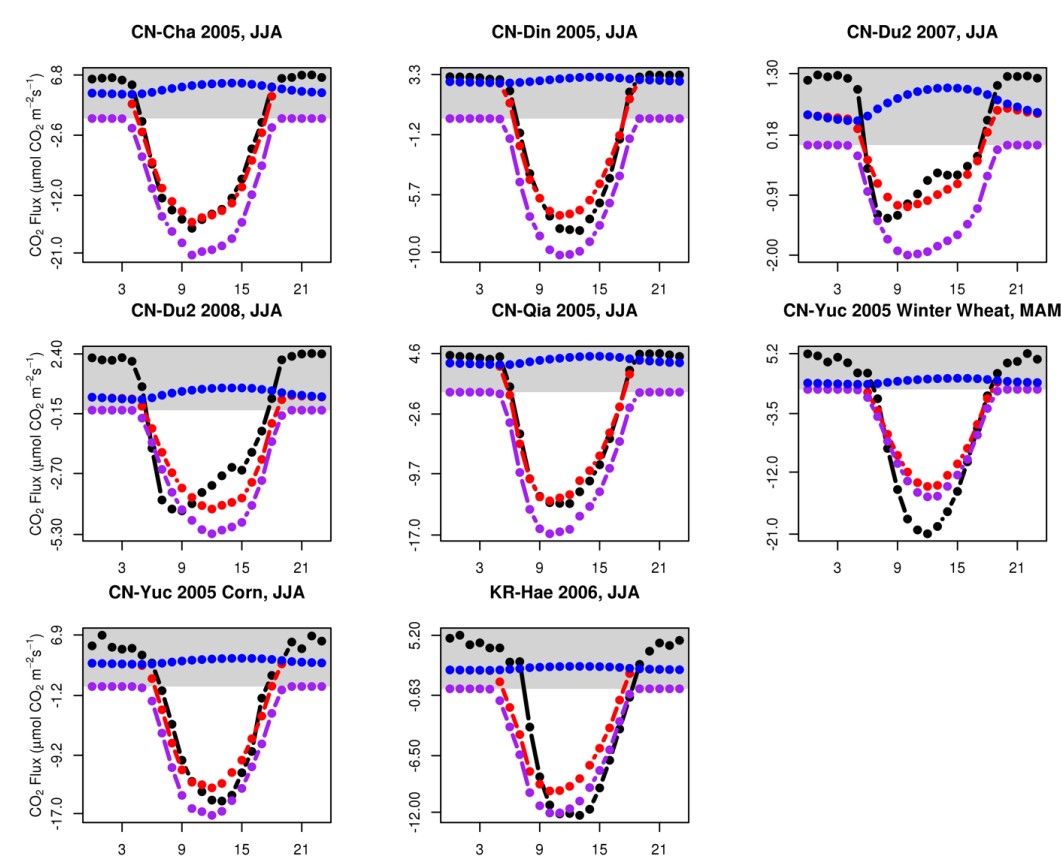

**Figure 5. Peak growing season diurnal mean measured *NEE* (black) along with modeled *NEE* (red), modeled *GPP* (purple), and modeled $R_{eco}$ (blue) vs. Local hour of day (UTC+8h). Grey shaded region represents region of positive fluxes (release to atmosphere).**





**Table 2. VPRM-CHINA scaling parameters by IGBP class compared to previous studies *Includes validation sites.**
**Units are: λ: μmolCO$_2$ m$^{-2}$s$^{-1}$(μmol PARm$^{-2}$s$^{-1}$)$^{-1}$; α: μmolCO$_2$ m$^{-2}$s$^{-1}$ °C$^{-1}$; β: μmolCO$_2$ m$^{-2}$s$^{-1}$; PAR$_o$: μmol m$^{-2}$s$^{-1}$**

| | | Mixed Forest | Evergreen Classes | Grassland* | Crops –Rice* | Woody Savanna | Crops – Corn | Crops – Winter Wheat |
|---|---|---|---|---|---|---|---|---|
| **This Study (East China)** | λ | 0.129 | 0.0903 | 0.0451 | 0.0583 | 0.104 | 0.143 | 0.157 |
| | α | 0.267 | 0.128 | 0.0306 | 0.0523 | 0.162 | 0.0938 | 0.0870 |
| | β | -0.291 | -0.464 | 0.0919 | -0.0769 | -0.710 | 1.42 | 0.00604 |
| | PAR$_0$ | 639 | 786 | 464 | 2030 | 1405 | 2070 | 1205 |
| **Hilton et al. (2013): North America** | λ | 0.102 | 0.107 | 0.0583 | -- | 0.0571 | 0.0826 | -- |
| | α | 0.249 | 0.110 | 0.0523 | -- | 2.33E-3 | 0.0510 | -- |
| | β | 7.18E-4 | 0.189 | -0.0769 | -- | 0.592 | 0.792 | -- |
| | PAR$_0$ | 565 | 777 | 2030 | -- | 3500 | 4173 | -- |
| **Mahadevan et al. (2008): North America** | λ | 0.123 | 0.114 | 0.213 | -- | 0.057 | 0.075 | -- |
| | α | 0.244 | 0.153 | 0.028 | -- | 0.012 | 0.173 | -- |
| | β | -0.24 | 1.56 | 0.72 | -- | 0.58 | 0.82 | -- |
| | PAR$_0$ | 629 | 790 | 542 | -- | 3241 | 11250 | -- |

We set our coordinate system such that uptake by the biosphere is represented as a negative *NEE* and release to the atmosphere is represented as a positive number. The VPRM-CHINA bias is typically less at hourly timescales (Fig. 5, Table 3), where processes are largely determined by temperature and light parameterizations. However, the unexplained variance is non-random and aggregates over longer timescales (e.g., months and years) and is evident in larger monthly mean biases as

in Fig. 6, and in annual sums (Table 4). On annual scales, mean annual temperature and precipitation have been found to dominate carbon exchange in the Asian region (Chen et al., 2013). However, the relationship between cumulative rainfall and *LSWI* varied depending on whether rainfall is in a high regime or a low regime (Chandrasekar et al., 2010). Future versions of VPRM-CHINA for China would benefit from using Solar Induced Fluorescence (SIF), a more reliable method for quantifying photosynthetic capacity that replaces the *EVI*, $P_{scale}$, and $W_{scale}$ terms from the VPRM-CHINA *NEE* equation

(Luus et al., 2017). The time period of this study does not coincide with SIF data availability. At monthly resolutions modeled uptake underestimates observed uptake during the growing season, particularly in the CN-Yuc cropland site during both the winter wheat and corn modes. For all sites and timescales, except evergreen broadleaf and woody savannas that have little influence on the receptor, respiration is underestimated by the model by an additive offset.




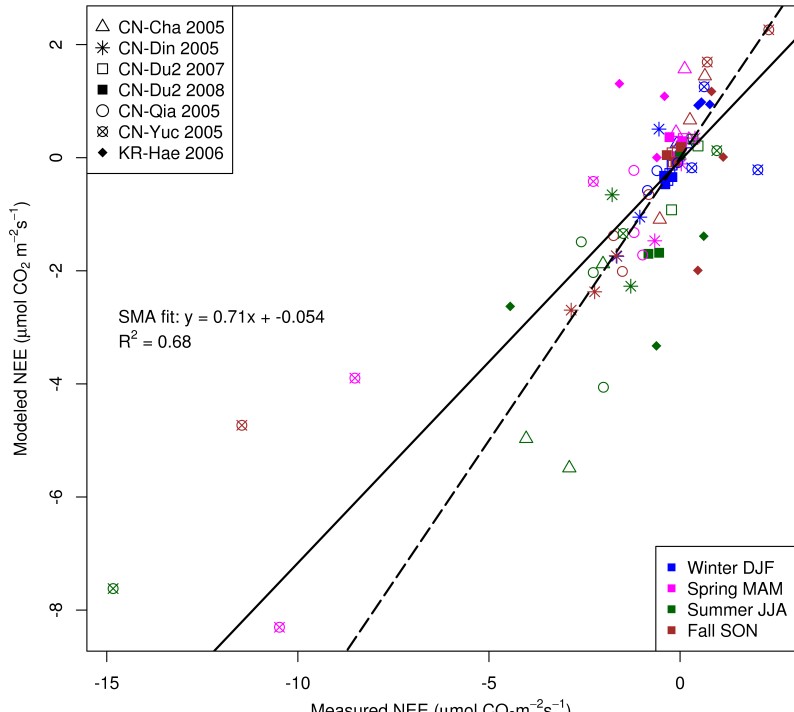

**Figure 6. Monthly means of predicted (Modeled) *NEE* vs Measured *NEE* at calibration sites, colored by season. Solid line is Standard Major Axis (SMA) regression line; dashed line is 1:1.**

5    **Table 3. Residual Standard Error and 1-σ values for each calibration site in study domain.**

|  | Mixed Forest | Evergreen Broadleaf | Grassland | Woody Savanna | Wheat/ Corn |
|---|---|---|---|---|---|
| RSE | 2.779 | 2.211 | .6567 | 2.641 | 5.556/ 7.387 |
| $\sigma(\lambda)$ | 0.003 | 0.002 | 0.002 | 0.002 | 0.009/ 0.008 |
| $\sigma(\alpha)$ | 0.005 | 0.004 | 0.001 | 0.004 | 0.018/ 0.020 |
| $\sigma(\beta)$ | 0.047 | 0.074 | 0.008 | 0.064 | 0.268/ 0.383 |
| $\sigma(PAR_0)$ | 21.61 | 25.68 | 34.81 | 52.15 | 149.5/ 340.1 |



### 3.3 Comparison to NASA CMS

We compare VPRM-CHINA performance at hourly timesteps to the NASA CMS weighted ensemble optimal mean of 15 vegetation models (Fisher et al., 2010). The CMS *NEE* is reported as fluxes of C, provided at 3-hourly resolution on a global 0.5º×0.5º grid. For direct comparison, we convert to fluxes of $CO_2$ and regrid the CMS using NCLv6.2.1 and a nearest-neighbor approach to the same spatial and temporal resolution and extent as the final VPRM-CHINA (hourly; 0.25º×0.25º). Figure 7 compares mean and 1-σ of annual uptake as kg C m$^{-2}$ y$^{-1}$ averaged over the 2005 to 2009 study time period. The CMS product exhibits minimal spatial heterogeneity relative to the VPRM-CHINA product.

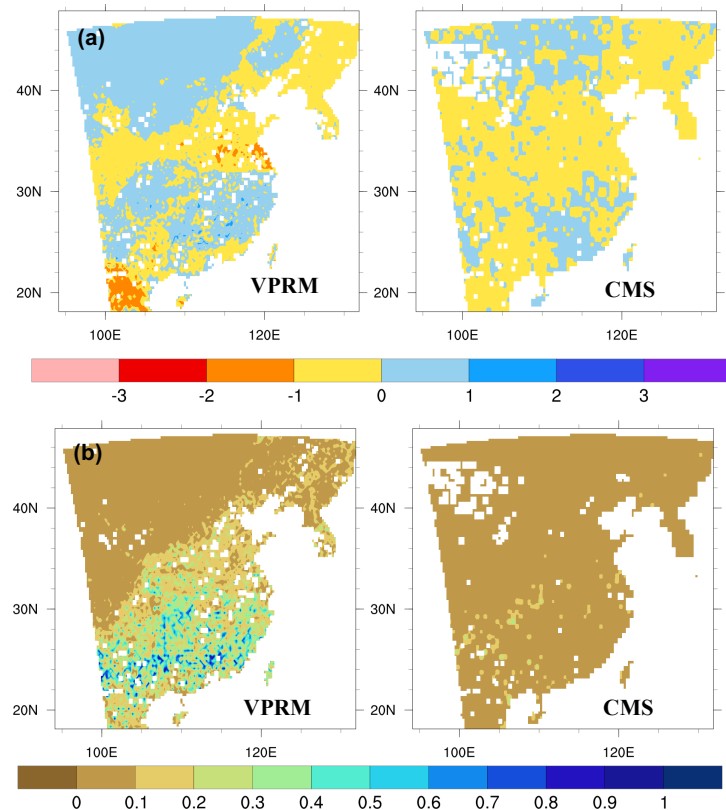

**Figure 7.** Annual *NEE* (kg C m$^{-2}$) modeled by VPRM-CHINA and CMS reported as multi-year (2005-2009) (a) means, and (b) 1-σ standard deviation



### 3.4 Multi-year Growing Season Analysis

Figure 8 displays modeled (ZHAO_VPRM, ZHAO_CMS) and measured $CO_2$ and residuals for each year during the growing season (May through September) incorporating the peak winter wheat period. We apply VPRM parameters obtained from the previously described calibration. Plots of daily averaged concentrations (Fig. 8a) suggest underestimated uptake by

5    CMS; further examination of modeled-measured residuals at the hourly scale in (Fig. 8b) indicate a systematic underestimate of *NEE* that is not present in the hourly $CO_2$ modeled with VPRM-CHINA *NEE*. Fig. 8c indicates the distributions of ZHAO_VPRM-CHINA $CO_2$ and measured $CO_2$ are similar; this is not the case with ZHAO_CMS during times of year where uptake is dominant.

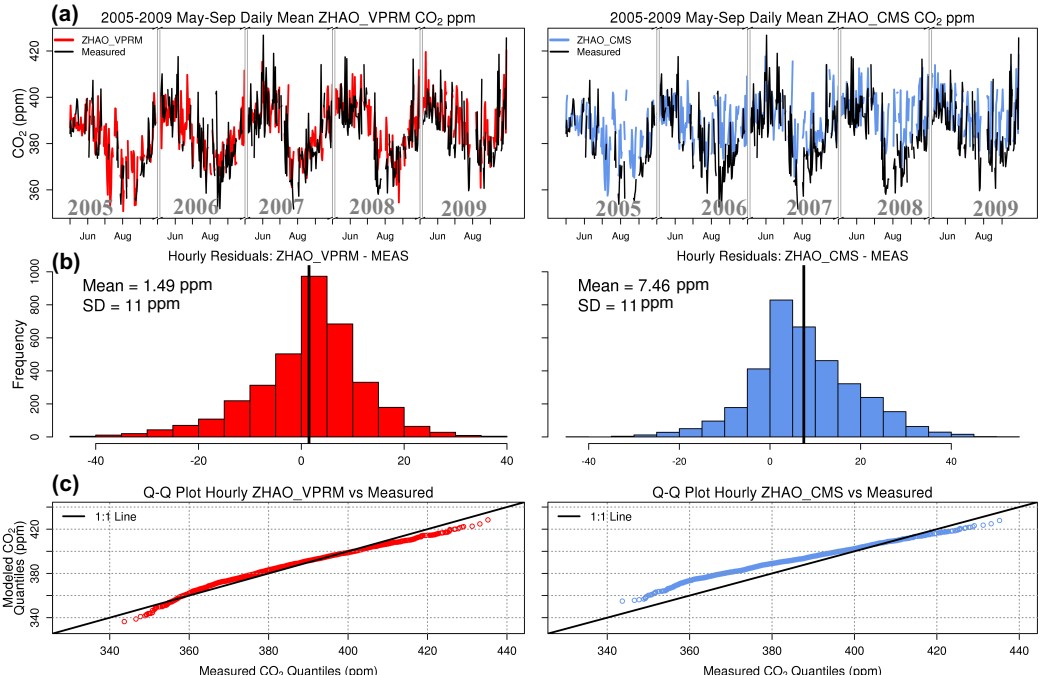

**Figure 8. Comparison of Modeled and Measured $CO_2$ during May-September 2005-2009, using two different vegetation models. (a) Daily Averages, for visual clarity; (b) Distribution hourly Modeled-Measured Residuals; (c) Comparison of modeled and measured hourly distributions using a Q-Q plot.**

10    We further examine the relative importance of the vegetation and anthropogenic influence by separately excluding each of the two main components (vegetation and anthropogenic) from the overall unoptimized modeled hourly $CO_2$ (Fig. 9).





We find that accurate modeling of growing season $CO_2$ in regions of China that are heavily influenced by anthropogenic activity requires incorporation of anthropogenic emissions. While unoptimized $CO_2$ concentrations modeled only by VPRM were in good agreement with data from aircraft surveys and tower sites in studies in North America (northern New England and Quebec, Matross et al., 2006), the relative magnitudes of the biological and anthropogenic fluxes in the eastern portion

5    of China are of comparable magnitude. In Fig. 9, we compare the ability of VPRM-CHINA and CMS to reproduce growing season $CO_2$ measurements using a fixed anthropogenic component prescribed by ZHAO. We find that when both vegetation and anthropogenic components are included the VPRM-CHINA vegetation fluxes result in a $CO_2$ prior that is less biased than one that uses CMS vegetation fluxes. We report mean bias, calculated as the average difference between modeled and measured $CO_2$. Based on results displayed in Fig. 8, we conclude that the apparent lower bias of CMS only in Fig. 9d is an

10   artefact of its lower prescription of biosphere uptake in the growing season.

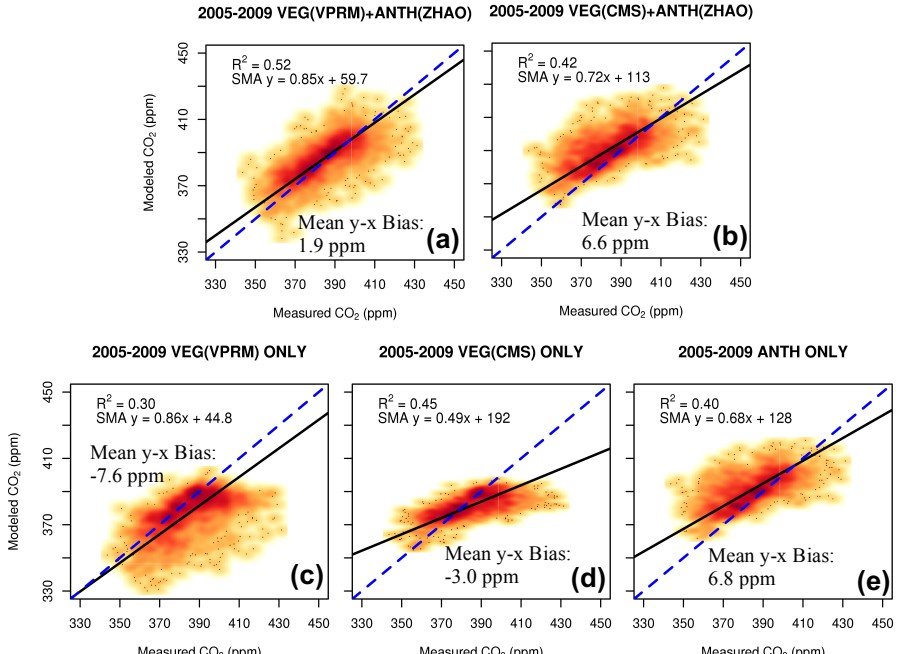

**Figure 9. Smoothed scatterplots examining relative impact of excluding anthropogenic and vegetation influences in modeling growing season $CO_2$. (a) Both hourly anthropogenic and VPRM-CHINA *NEE* included; (b) both hourly anthropogenic and 3-hourly CMS *NEE*; (c) hourly VPRM-CHINA *NEE* only; (d) 3-hourly CMS *NEE* only; (e) hourly anthropogenic only. All modeled inventories are unoptimized. The blue dashed line represents the 1:1 line; the solid black line represents the SMA fit line described by the equations displayed. Higher point density is represented by darker colors.**



Figure 10 displays modeled anthropogenic (ZHAO) and vegetation (VPRM-CHINA) contributions to the measured $CO_2$ signal relative to background concentrations during the May-September growing season. Peak drawdown occurs in August. Extending the results from Fig. 9c and Fig. 9e, we model comparable magnitudes of anthropogenic emissions and biological uptake particularly during the peak growing season.

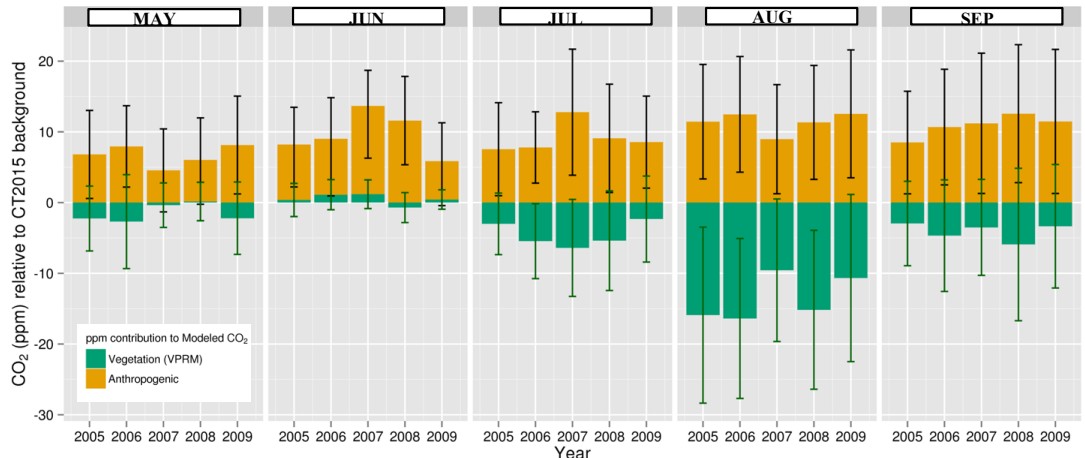

**Figure 10. Modeled mean monthly contribution (ppm) to Miyun $CO_2$ from vegetation (VPRM-CHINA) and anthropogenic (ZHAO) sources, relative to advected CT2015 background concentrations during the regional growing season (MJJAS). Error bars represent 1-σ of monthly averages (Green: VPRM-CHINA Vegetation; Black: ZHAO Anthropogenic). Negative values represent depletion from CT2015 background; positive values represent enhancement.**

### 3.5 Performance of VPRM-CHINA on Annual Timescales

We stress that the VPRM-CHINA is primarily intended as an hourly prior, capturing $CO_2$ flux covariance with spatial and temporal patterns in vegetation type and status detected by remote sensing. However, analysis at annual timescales is useful to illustrate regional biases. As such, we present analysis of VPRM-CHINA on annual timescales over the study time period

10  by comparison to CMS and Piao et al. (2009). Piao et al. (2009) divided China into nine main regions roughly corresponding to China's administrative regions and examined the vegetation carbon budget on annual timescales using models and vegetation products spanning the time period from 1980 through 2005.

We break down our study region similarly to Piao et al. (2009) as shown in Fig. 11; our study region includes seven of

15  Piao's nine regions. In addition, we calculate the multi-year (2005-2009) mean annual STILT footprints in the study domain which provides an estimate of the regions that have the greatest influence on the ultimate signal at the Miyun receptor. These




regions are displayed with contour lines of 50th, 75th, and 90th percentiles of surface footprints. North China, Inner Mongolia, and Northeast China are the administrative regions significantly encompassed by the mean annual 90th percentile region; it follows that parameterizations of vegetation and anthropogenic $CO_2$ fluxes in these regions have the greatest impact on the modeled $CO_2$ signal.

We examine VPRM-CHINA performance in each of these three administrative regions relative to other vegetation models (Table 4). Consistent with Piao et al. (2009), we do not include croplands in our annual totals in Table 4 due to the rapid turnover of cropland carbon stocks. We find that across the administrative regions encompassed by the STILT influence contours, there is agreement within uncertainty across models. However, it is important to note that Piao et al. (2008) include

10    estimates from a significantly earlier time period (1980 to 2005). Regionally, there is agreement across all models within uncertainty bounds. On a sub-region basis, there is agreement across models in the direction of carbon flux with the exception of Inner Mongolia.

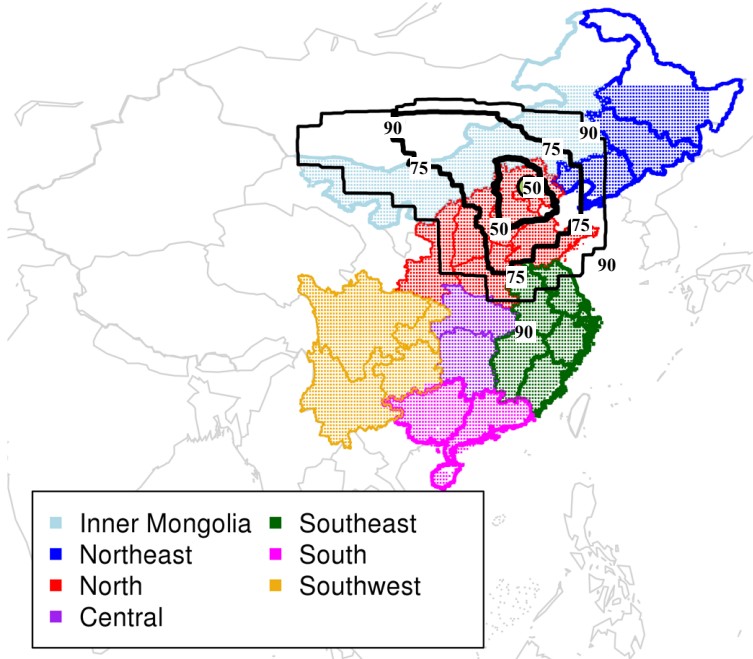

**Figure 11. STILT influences and major administrative regions of China in study domain. Administrative regions are categorized according to convention in Piao et al. (2009). Stippling represents location of 0.25°x0.25° VPRM-CHINA and ZHAO gridcell centers within the regions. Black contour lines display 50th, 75th, and 90th percentiles of mean annual STILT footprints (2005-2009) to highlight regions with the greatest influence on Miyun receptor (solid green circle). Note that two of the regions (Inner Mongolia and Northeast China) are mostly, but not completely, encompassed by our study domain.**



**Table 4.** Comparison of unoptimized VPRM-CHINA annual carbon exchange by region of China with other vegetation models. VPRM-CHINA and CMS values are reported as multi-year means±2σ. Reported Piao values are the average of three approaches (process-based models, inverse approach, inventory approach). [a]VPRM-CHINA and regridded CMS do not cover extent of region; [b]Does not include land use less than 5%; [c]Cropland extent included for reference; but are not included in regional budget due to rapid turnover of carbon stocks, consistent with Piao et al., 2008. Negative quantities: uptake by the biosphere.

| Region | MODIS IGBP Dominant Categories (% prevalence)[b] | VPRM-CHINA (2005-2009) TgCyr[-1] | CMS (2005-2009) TgCyr[-1] | Piao (1980-2005) TgCyr[-1] |
|---|---|---|---|---|
| Inner Mongolia[a] | Grasslands (61%) Croplands (5%)[c] Grasslands/Barren (29%) | 64±7 | -2.6±16 | -8.3±11 |
| Northeast China[a] | Decid. Broadleaf (10%) Mix Forest (20%) Grasslands (10%) Croplands (47%)[c] Grasslands/Mosaic (9%) | -64±16 | -7.7±16 | -3.3±13 |
| North China | Mix Forest (13%) Grasslands (23%) Croplands (51%)[c] | -13±24 | -7.1±22 | -25.7±23 |
| Regional TOTAL | | -13±15 | -17±16 | -37±29 |

The VPRM-CHINA is poorly constrained by data in Inner Mongolia where we are restricted by the availability of data and use a single degraded grassland site (CN-Du2) to represent over 60% of its landscape. This is particularly problematic as grasslands are shown to have significant ecosystem variation (Zhang et al., 2014). In general, the lack of spatial and, at second order, temporal heterogeneity in both calibration and validation data for the VPRM-CHINA lead to significant error propagation at annual timescales. Jiang et al. (2016) further illustrate this point by evaluating the effect of assimilating more measurements on carbon sink estimates. Jiang et al. (2016) present top-down estimates of land carbon sinks using Carbon Tracker China (CTC) and a Bayesian Inversion (BI) for all of China, noting that the observation network density is highest in the north and east and therefore biased toward constraining exchange in these regions. Increasing the observational constraints from one station to three stations and aircraft data increase carbon sink estimates for the BI and CTC systems by 76% and 95%, respectively (Jiang et al., 2016) and reduce uncertainty in all cases.

In contrast, the VPRM-CHINA is best constrained by data in North and Northeast China. The land categories in these regions are appropriately represented by their respective eddy flux calibration sites. For instance, North China has a high prevalence of heavily disturbed grasslands appropriately represented by CN-Du2. In addition, mixed forests in North China



are well represented by CN-Cha, which is in close proximity. North and Northeast China fall completely within the 75th percentile of STILT footprints (Fig. 11), thereby increasing confidence in modeled $CO_2$ during the growing season. Similar to Zhang et al. (2014) we find a strong carbon sink in the northeast (Table 1-4), a feature not found by Piao et al. (2009).

## 4 Summary and Conclusions

This study shows how the VPRM can be adapted for use in large-scale $CO_2$ studies in China by (1) streamlining processing of driver datasets at reasonable computational timescales; (2) successfully addressing the significant extent of dual-cropped regions in the North China Plain; (3) calibrating VPRM-CHINA with China-specific observations at the ecosystem level; and (4) demonstrating the low bias of VPRM-CHINA with respect to hourly growing-season $CO_2$ measured over multiple years at Miyun. We provide hourly estimates of *NEE*, *GPP*, and R for the eastern half of China from 2005-2009 on a

0.25°x0.25° grid. We assess its performance in regions that significantly influence $CO_2$ measured at the Miyun station, and compare it to modeled *NEE* from other vegetation $CO_2$ studies across China (NASA CMS, Piao et al., 2009). We use the ZHAO inventory as a control for anthropogenic $CO_2$ emissions within a WRFv3.6.1-STILT modeling framework.

  The VPRM-CHINA is calibrated with eddy flux data from each major IGBP ecosystem class in the domain (Fig. 1, Table 1).

We separately prescribe a winter wheat mode and corn mode for dual cropland classes within the North China Plain belt (Fig. 2). We find the *PAR* estimated from WRFv3.6.1 downward shortwave radiation to be overestimated relative to observations by a factor of 1.5 to 2, depending on season (Fig. 4). We find it necessary to scale *PAR* in Eq. 1 by the seasonal scaling factors for the regional VPRM-CHINA to realistically represent hourly and annual ecosystem carbon fluxes. Overall, the VPRM-CHINA calibration parameters obtained for this study agree with previous studies for similar ecosystem types at

similar latitude (Table 2). We find uptake to be well constrained at hourly scales (Fig. 5, Fig. 8, Fig. 9) but underestimated relative to observations at monthly scales (Fig. 6).

  We find greater spatial and temporal heterogeneity in VPRM-CHINA relative to CMS over the study time period (Fig. 7). Accounting for spatial patterns in carbon exchange is necessary for the Lagrangian transport model to capture effects on

observed $CO_2$ at fine grid scales. Indeed, at the hourly scale, the VPRM-CHINA shows significantly greater ability to capture vegetation processes influencing observations at Miyun relative to CMS (Fig. 8, Fig. 9). Due to Miyun's proximity to the North China Plain cropping region, cropland signatures strongly influence partitioning of contributions to modeled $CO_2$ enhancements relative to CT2015 background concentrations, as evidenced by the strongest drawdown occurring during the peak period of the corn-growing season (Fig. 10).


  The VPRM-CHINA is well constrained at all timescales by observational ecosystem data in Northeast, North, and South China but has sparse representation of its major ecosystem classes in Inner Mongolia, Central, Southeast and Southwest



China. Given that the major regions influencing observations at Miyun are in the North, Northeast and Inner Mongolia (Fig. 11) use of the VPRM-CHINA is appropriate for northern China sites. However, improvements in modeling growing season $CO_2$ for Miyun rely heavily on improved parameterizations of heterogeneous grasslands. At the annual scale, the VPRM-CHINA agrees with CMS and Piao et al. (2009) within uncertainty bounds in the 90th percentile contour of the influence region (Table 4).

Overall, the VPRM-CHINA performs well on multiple timescales (hourly to annually) in regions where it is constrained by representative ecosystem observations, stressing the importance of a dense observation network for larger scale studies influenced by other regions of China. In addition to more eddy flux ecosystem-level data, future versions of the VPRM-CHINA for China will benefit considerably from improvements in *PAR* fields; we are currently extrapolating *PAR* data using scaling factors derived from observations at five sites to the entire domain. The VPRM-CHINA is particularly appropriate for hourly and daily resolution timescales, which are the most relevant scales needed for use as a prior in signal attribution studies. Processes at these timescales are primarily driven by variations in temperature and *PAR*. In contrast, CMS performs poorly in resolving questions requiring hourly timescale processes. In addition to studies involving ground-based measurements, our results show that high resolution vegetation flux models such as VPRM-CHINA are critical for interpreting retrievals from global $CO_2$ remote sensing efforts such as the Orbiting Carbon Observatory (OCO) missions OCO-2 and OCO-3 (planned). Depending on satellite time-of-day and season of crossover, efforts to interpret the relative contribution of the vegetation and anthropogenic components to the measured signal are critical in key emitting regions such as Northern China--where the magnitude of the vegetation $CO_2$ signal is shown to be equivalent to the anthropogenic signal.



**Code and Data Availability**

Code and data are available at http://dx.doi.org/10.7910/DVN/RQLGLH. The supplement includes the VPRM-CHINA methods paper, relevant model code, calibration results, WRF temperature and shortwave radiation fields, and hourly 0.25°x0.25° modeled NEE and GPP.

**Author Contributions**

AD prepared the manuscript with contributions from JWM, SCW, TN, YW, CN, and MBM. YW is the PI of the Miyun dataset; YW and JWM provided access to the final quality-controlled hourly $CO_2$ observational data. YZ provided the anthropogenic inventory. AD developed VPRM-CHINA with assistance from KL. WRF-STILT simulations were performed by AD with assistance from TN.

**Competing Interests**

The authors declare no competing interests.

**Acknowledgments**

We thank the Harvard-China Project and the Harvard Global Institute for funding this study; Zhiming Kuang for providing computational resources; Lu Hu, Jialin Liu, and Jenna Samra for helpful discussion; Mary Haley for assistance with NCL;
and Shiping Chen and Haijun Peng for providing assistance with eddy flux data from CN-Du2 and CN-Yuc, respectively. This work used eddy covariance data acquired and shared by the FLUXNET community, including these networks: AmeriFlux, AfriFlux, AsiaFlux, CarboAfrica, CarboEuropeIP, CarboItaly, CarboMont, ChinaFlux, Fluxnet-Canada, GreenGrass, ICOS, KoFlux, LBA, NECC, OzFlux-TERN, TCOS-Siberia, and USCCC. The ERA-Interim reanalysis data are provided by ECMWF and processed by LSCE. The FLUXNET eddy covariance data processing and harmonization was
carried out by the European Fluxes Database Cluster, AmeriFlux Management Project, and Fluxdata project of FLUXNET, with the support of CDIAC and ICOS Ecosystem Thematic Center, and the OzFlux, ChinaFlux and AsiaFlux offices. CarbonTracker CT2015 results provided by NOAA/ESRL, Boulder, Colorado, USA, http://carbontracker.noaa.gov.



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
