# Peer review of "Assessing biotic contributions to CO2 fluxes in Northern China using the Vegetation, Photosynthesis and Respiration Model (VPRM-CHINA) and observations from 2005 to 2009."

_Biogeosciences, 2017_

## Referee Comment (RC1) · Anonymous Referee #1 · 11 Feb 2018

The paper describes the first implementation of the diagnostic CO2 flux model VPRM for a large part of China. In combination with gridded emission inventory data for CO2 and atmospheric transport simulations with WRF-STILT, hourly atmospheric model fractions are simulated and compared to observations made at a rural site in northern China. The manuscript is well written, with clear tables and figures, and fits well within the scope of the journal. I recommend accepting the manuscript after applying a few minor changes mentioned in my comments below.

[Figure]

Main comments:

I have a concern when setting NEE to missing values for shrubland vegetation classes, as this would mean that all simulated $CO_2$ values containing influence from that vegetation class are missing as well. How is this handled in the model? Would it not be better to either set NEE in those cases to zero, or to a value that is somewhat in the range of the observed fluxes?

Specific comments:

P3 L29: please clarify what is meant by "hourly $CO_2$ observations". I assume atmospheric mole fractions have been measured.

Eq. 4a: LSWI_max should be described.

Fig. 4a: please mention what the colors (read, black) indicate

P19 L11: please explain in more detail what is meant by "unoptimized"

---

## Referee Comment (RC2) · Anonymous Referee #2 · 31 Jul 2018

General commentsïijŽThis manuscript reported on parameters adapting and model diagnosing of VPRM-CHINA for the eastern half of China.This paper is a well-presented and scientifically sound study and I recommend it for publication after minor revisions. However, I think the title '...Using the Vegetation, Photosynthesis, and Respiration Model to partition contributions to CO2 measurements ...' is not very proper for this paper: VPRM-CHINA model isn't able to partition contributions of CO2 concerntration. A more scientific titile should be needed for this manuscript .

[Figure]

Specific comments: P9 Table 1 Please explain clearly what is meant by 'CN' or 'KR'). P11 L11 Please describe how to scale scale annual gridcell emissions (as Gg CO2) to $\mu$molCO2 m-2s-1

---

## Author Comment (AC1) · 7 Aug 2018

(RC 1) I have a concern when setting NEE to missing values for shrubland vegetation classes, as this would mean that all simulated CO2 values containing influence from that vegetation class are missing as well. How is this handled in the model? Would it not be better to either set NEE in those cases to zero, or to a value that is somewhat in the range of the observed fluxes?

[Figure]

(AC 1) Thank you for the comment. Shrublands constitute less than 1.5% of the domain land use area and do not appreciably affect the modeled atmospheric $CO_2$ values. The 7-day-backtrajectory vegetation effect in ppm for each measurement hour is the result of a spatiotemporal sum of footprint*fluxes – as such, setting the NEE for these pixels to 0 is numerically the same as setting the pixels to missing. In order to estimate the influence on the $CO_2$ mixing ratio at Miyun from the shrubland (IGBP-6, IGBP-7), needleleaf (IGBP-1, IGBP-3), and permanent wetland (IGBP-11) land classes, we compare the sum of surface influence from pixels corresponding to those land categories to the total average (2005 to 2009) annual STILT surface influence. We find that these five land classes only account for 4% of the total influence mostly from the two the shrubland classes. The croplands, grasslands and mixed forests that are best represented in the VPRM training data account for 86% of the total influence.

We have edited the text in the manuscript to make this justification clearer.

Specifically:

P6 L13: "Pixels corresponding to these ecosystem types have NEE values set to missing. We justify this assumption in Sect. 3."

P14 L1 to L9: "As noted in Sect. 2.2.2, NEE values for shrubland ecosystems are set to missing. The vegetation effect on $CO_2$ in ppm for each measurement hour is the result of a spatiotemporal sum of the product of the STILT footprint and surface fluxes. As such, an NEE pixel of 'missing' is numerically identical to an NEE pixel set to zero. Our choice to set these values as missing is based on the reasoning that a zero value (or a previously published value that has low confidence) implies that we know more about these shrubland ecosystems than we do in this domain. By comparing the sum of surface influence from shrubland, needleleaf, and permanent wetland ecosystems to the total average annual surface influence, we find these ecosystems contribute less than 5% to the total influence. As such, setting these classes to missing does not appreciably affect the conclusions."
* * *
(RC 2) 1. P3 L29: please clarify what is meant by "hourly $CO_2$ observations". I assume atmospheric mole fractions have been measured.

(AC 2) We have replaced this wording (P3 L29): "We evaluate performance of the VPRM-CHINA during the growing season using five years (2005-2009) of hourly averages of continuously measured $CO_2$ (LI-COR Biosciences Li-7000)."
* * *
(RC 3) 1. Eq. 4a: LSWI_max should be described.

(AC 3) Thank you, we have edited the text to clarify this (P5 L14):

"Wscale is derived from both LSWI and the maximum LSWI (LSWImax) for a particular growing season"
* * *
(RC 4) Fig. 4a: please mention what the colors (read, black) indicate

(AC 4) Thank you for noting this. We have edited accordingly: "Aggregated mean modeled (red) and measured (black) PAR for each eddy flux calibration site by season"
* * *
(RC 5) P19 L11: please explain in more detail what is meant by "unoptimized"

(AC 5) We have clarified our usage of this term (now P20 L10):

"We further examine the relative importance of the vegetation and anthropogenic influence by separately excluding each of the vegetation and anthropogenic components from the overall unoptimized (i.e., inventories uncorrected by observations) modeled hourly $CO_2$ (Fig. 9)."

---

## Author Comment (AC2) · 7 Aug 2018

(RC 1) This manuscript reported on parameters adapting and model diagnosing of VPRM-CHINA for the eastern half of China.This paper is a well-presented and scientifically sound study and I recommend it for publication after minor revisions. However, I think the title '...Using the Vegetation, Photosynthesis, and Respiration Model to partition contributions to CO2 measurements ...' is not very proper for this paper:

[Figure]

VPRM-CHINA model isn't able to partition contributions of CO2 concerntration. A more scientific titile should be needed for this manuscript .

(AC 1) Thank you for pointing this out. We emphasize that this is an effort to *model* partitioning. The VPRM is a helpful tool to provide insight into modeled estimates of what the relative contributions of vegetation and anthropogenic activity are to the atmospheric signal. We explore this concept in more detail in Section 3.4 (Fig. 9 and Fig. 10). As you point out, we cannot partition this in the real world, but we are showing that the VPRM-China model can do so to a certain extent and one of the purposes of the paper is to justify its use for this purpose by evaluating the vegetation-dominated growing season modeled time series relative to observations. We are unfortunately restricted by eddy flux site data availability – the uncertainty associated with this early version of VPRM-China (and, by extension, this study's efforts to model partitioning) would be considerably reduced were additional validation data available. That being said, we are certainly open to a more appropriate title, if the editor or referee has suggestions.

—————

(RC 2) P9 Table 1 Please explain clearly what is meant by 'CN' or 'KR').

(AC 2) This has now been clarified in the Table 1 caption: "Site Name abbreviations are according to FLUXNET convention; CN=China; KR=South Korea."

—————

(RC 3) P11 L11 Please describe how to scale scale annual gridcell emissions (as Gg CO2) to $\mu$molCO2 m-2s-1

(AC 3) We have now clarified this in the text (P11 L11 to L13): Gridded annual emissions (Gg CO2) are converted to fluxes in umol/m2/s by dividing the annual emission by area in the grid cell and number of seconds in the year.

---

## Author Response (AR1)

Author responses to Anonymous Referee # 1
Referee comments in boldface, author comments in normal typeface.

1. **I have a concern when setting NEE to missing values for shrubland vegetation classes, as this would mean that all simulated CO2 values containing influence from that vegetation class are missing as well. How is this handled in the model? Would it not be better to either set NEE in those cases to zero, or to a value that is somewhat in the range of the observed fluxes?**

   Thank you for the comment. Shrublands constitute less than 1.5% of the domain land use area and do not appreciably affect the modeled atmospheric $CO_2$ values. The 7-day-backtrajectory vegetation effect in ppm for each measurement hour is the result of a spatiotemporal sum of footprint*fluxes -- as such, setting the NEE for these pixels to 0 is numerically the same as setting the pixels to missing. In order to estimate the influence on the CO2 mixing ratio at Miyun from the shrubland (IGBP-6, IGBP-7), needleleaf (IGBP-1, IGBP-3), and permanent wetland (IGBP-11) land classes, we compare the sum of surface influence from pixels corresponding to those land categories to the total average (2005 to 2009) annual STILT surface influence. We find that these five land classes only account for 4% of the total influence mostly from the two the shrubland classes. The croplands, grasslands and mixed forests that are best represented in the VPRM training data account for 86% of the total influence.

   We have edited the text in the manuscript to make this justification clearer.

   Specifically:

   P6 L13: "Pixels corresponding to these ecosystem types have NEE values set to missing. We justify this assumption in Sect. 3."

   P14 L1 to L9: "As noted in Sect. 2.2.2, NEE values for shrubland ecosystems are set to missing. The vegetation effect on $CO_2$ in ppm for each measurement hour is the result of a spatiotemporal sum of the product of the STILT footprint and surface fluxes. As such, an NEE pixel of 'missing' is numerically identical to an NEE pixel set to zero. Our choice to set these values as missing is based on the reasoning that a zero value (or a previously published value that has low confidence) implies that we know more about these shrubland ecosystems than we do in this domain. By comparing the sum of surface influence from shrubland, needleleaf, and permanent wetland ecosystems to the total average annual surface influence, we find these ecosystems contribute less than 5% to the total influence. As such, setting these classes to missing does not appreciably affect the conclusions."

2. **P3 L29: please clarify what is meant by "hourly CO2 observations". I assume atmospheric mole fractions have been measured.**

   We have replaced this wording (P3 L29):

"We evaluate performance of the VPRM-CHINA during the growing season using five years (2005-2009) of hourly averages of continuously measured CO2 (LI-COR Biosciences Li-7000)."

3. **Eq. 4a: LSWI_max should be described.**

Thank you, we have edited the text to clarify this (P5 L14):

"Wscale is derived from both LSWI and the maximum LSWI (LSWImax) for a particular growing season"

4. **Fig. 4a: please mention what the colors (read, black) indicate**

Thank you for noting this. We have edited accordingly:
"Aggregated mean modeled (red) and measured (black) PAR for each eddy flux calibration site by season"

5. **P19 L11: please explain in more detail what is meant by "unoptimized"**

We have clarified our usage of this term (now P20 L10):

"We further examine the relative importance of the vegetation and anthropogenic influence by separately excluding each of the vegetation and anthropogenic components from the overall unoptimized (i.e., inventories uncorrected by observations) modeled hourly CO2 (Fig. 9)."

Author responses to Anonymous Referee # 2
Referee comments in boldface, author comments in normal typeface.

1. **This manuscript reported on parameters adapting and model diagnosing of VPRM-CHINA for the eastern half of China.This paper is a well-presented and scientifically sound study and I recommend it for publication after minor revisions. However, I think the title '...Using the Vegetation, Photosynthesis, and Respiration Model to partition contributions to CO2 measurements ...' is not very proper for this paper: VPRM-CHINA model isn't able to partition contributions of CO2 concerntration. A more scientific titile should be needed for this manuscript .**

   Thank you for pointing this out. We emphasize that this is an effort to model partitioning. The VPRM is a helpful tool to provide insight into **modeled estimates** of what the relative contributions of vegetation and anthropogenic activity are to the atmospheric signal. We explore this concept in more detail in Section 3.4 (Fig. 9 and Fig. 10). As you point out, we cannot partition this in the real world, but we are showing that the VPRM-China model can do so to a certain extent and one of the purposes of the paper is to justify its use for this purpose by evaluating the vegetation-dominated growing season modeled time series relative to observations. We are unfortunately restricted by eddy flux site data availability – the uncertainty associated with this early version of VPRM-China (and, by extension, this study's efforts to model partitioning) would be considerably reduced were additional validation data available. That being said, we are certainly open to a more appropriate title, if the editor or referee has suggestions.

2. **P9 Table 1 Please explain clearly what is meant by 'CN' or 'KR').**

   This has now been clarified in the Table 1 caption:
   "Site Name abbreviations are according to FLUXNET convention; CN=China; KR=South Korea."

3. **P11 L11 Please describe how to scale scale annual gridcell emissions (as Gg CO2) to μmolCO2 m-2s-1**

   We have now clarified this in the text (P11 L11 to L13):
   "We therefore directly scale annual gridcell emissions (as Gg CO2) to $\mu molCO_2$ $m^{-2}s^{-1}$ by scaling (1) by area per gridcell computed as a function of latitude and (2) temporally from an annual time base to a seconds time base."

[revised manuscript text omitted]